# Effect of the Aqueous Extract of *Chrysobalanus icaco* Leaves on Maternal Reproductive Outcomes and Fetal Development in Wistar Rats

Natalie Emanuelle Ribeiro Rodrigues [1,2,*], Alisson Rodrigo da Silva Oliveira [1], Sandrine Maria de Arruda Lima [1], Daniel Medeiros Nunes [2], Priscilla Barbosa Sales de Albuquerque [2], Maria das Graças Carneiro da Cunha [3], Almir Gonçalves Wanderley [4,5], Flavio Manoel Rodrigues da Silva Júnior [6], José Bruno Nunes Ferreira Silva [7], Álvaro Aguiar Coelho Teixeira [8] and Teresinha Gonçalves da Silva [1,*]

[1] Laboratory of Farmatoxicological Prospecting of Bioactive Products (BIOFARMATOX), Department of Antibiotics, Federal University of Pernambuco (UFPE), Recife 54740-520, Pernambuco, Brazil; alissonrodrigoo@hotmail.com (A.R.d.S.O.); sandrinearruda@gmail.com (S.M.d.A.L.)

[2] Department of Medicine, University of Pernambuco (UPE), Garanhuns 53294-902, Pernambuco, Brazil; daniel.medeirosnunes@upe.br (D.M.N.); priscilla.albuquerque@upe.br (P.B.S.d.A.)

[3] Biochemistry Department, Federal University of Pernambuco (UFPE), Recife 50670-420, Pernambuco, Brazil; mgcc@ufpe.br

[4] Department of Physiology and Pharmacology, Federal University of Pernambuco (UFPE), Recife 50670-901, Pernambuco, Brazil; almir.wanderley@unifesp.br

[5] Department of Pharmaceutical Sciences, Federal Univesity of São Paulo, São Paulo 09913-030, Brazil

[6] Institute of Biological Sciences, Federal University of Rio Grande-FURG, Rio Grande 96293-900, Rio Grande do Sul, Brazil; f.m.r.silvajunior@gmail.com

[7] Laboratory of Biotechnology, Immunology and Health Studies, Medicine Course, Federal University of Tocantins (UFT), Palmas 77001-923, Tocantins, Brazil; nunes.brj@mail.uft.edu.br

[8] Department of Morphology and Animal Physiology, Federal Rural University of Pernambuco (UFPE), Recife 52171-900, Pernambuco, Brazil; alvaro@dmfa.ufrpe.br

* Correspondence: natalie.rodrigues@upe.br (N.E.R.R.); teresinha.goncalves@ufpe.br (T.G.d.S.)

**Abstract:** Toxicological studies on medicinal plants are essential to ensure their safety and effectiveness in treating various diseases. Despite the species *Chrysobalanus icaco* L. being popularly used in the treatment of several diseases due to the pharmacological properties of its bioactive compounds, there are few studies in the literature regarding its toxicity regarding reproduction. Therefore, the purpose of this study was to assess the potential embryotoxic and teratogenic effects of the aqueous extract of *C. icaco* leaves (AECi) on Wistar rats. Animals were given AECi at doses of 100, 200, and 400 mg/kg during the pre-implantation and organogenesis periods. Data were analyzed using ANOVA followed by Tukey's test and Kruskal–Wallis. Pregnant rats treated during the pre-implantation period showed no signs of reproductive toxicity. Rats that received AECi at 100, 200, and 400 mg/kg during organogenesis did not exhibit any signs of maternal systemic toxicity or significant differences in gestational and embryotoxic parameters. Some skeletal changes were observed in the treated groups. Therefore, it can be suggested that AECi at doses of 100, 200, and 400 mg/kg is safe for treated animals and does not induce reproductive toxicity under the experimental conditions applied, but it also caused low systemic toxicity.

**Keywords:** toxicity; *Chrysobalanus icaco*; embryogenesis; organogenesis

## 1. Introduction

During the prenatal and postnatal period, exposure to factors such as stress, food deprivation, insecticides, chemicals, and disease can affect fetal development and lead to birth defects in neonatals [1–4].

Phytotherapy is used worldwide as a therapeutic resource by various populations, including pregnant women. However, most plants with phytotherapeutic potential do not

receive adequate attention related to their toxicity, i.e., for most of them, there are no data about their safety of use, especially during pregnancy [5,6].

Several tests have been used to evaluate the potential toxicity of substances since the 16th century. However, in the 1990s, the use of animals in toxicological tests increased exponentially, especially after the establishment of the lethal dose protocol of 50% ($DL_{50}$). After that, regression analyses were developed to adjust the dose–response effects in animals, aiming at the calculation of the $LD_{50}$. During the 20th century, several other methods of toxicological tests using laboratory animals were developed, many of which are still used today, such as embryotoxicity tests. For more than 40 years of development, the methodologies of toxicological evaluations have remained relatively unchanged and cover many well-known and widely used models in the scientific community [7].

*Chrysobalanus icaco* L., belonging to the Chrysobalanaceae family and popularly known as "Abajuru", "Guajuru", and "Ajuru", among others, is native to both America and Africa [8]. In the Americas, *Chrysobalanus icaco* L. is distributed mainly along coastal areas from Florida (USA) to southern Brazil [9]. It is a medium-sized bushy plant, with alternate and simple leaves; pentamerous flowers and usually white or purple; and dry fruit or fleshy drupe and planoconvex cotyledons [10]. Riverside populations consumed its fruit in natura [11], while its roots and leaves are used in traditional medicine in the form of tea, especially for the treatment of diseases such as hemorrhage and chronic diarrhea [12].

Tests in rodents have reported that the aqueous extract of the bark and leaves of *C. icaco* have analgesic properties [13,14]. Other pharmacological studies have also been reported, for example, on the anti-inflammatory activity of the aqueous extract of its bark [13], the antimicrobial activity of the methanolic leaf extract, and the hypoglycemic activity of the aqueous extract of its leaves [15,16]. Additionally, the aqueous extract of *C. icaco* leaves can prevent weight gain and liver fat accumulation in hypercaloric diet-induced obese mice [17,18].

Despite the already published pharmacological effects and traditional use of *C. icaco*, to our knowledge, there are no commercial products based on this plant; in addition, the reports of its toxicological potential are limited, especially on the interferences in the animal reproductive period. Some studies have shown that the administration of the aqueous extract of leaves and bark of *C. icaco* does not induce signs of toxicity at a dose of 2000 mg/kg [13,19]. However, Ribeiro et al. [19] (2020) demonstrated that the administration of the aqueous extract of *C. icaco* leaves at doses of 100, 200, and 400 mg/kg, over 28 days, caused mild hepatic and renal toxicity.

The search for new products of natural origin that benefit the health and well-being of the population has grown in recent years. Furthermore, the development of novel forms of treatment not only contributes to strengthening national science through the accumulation of know-how, but also to building a sustainable economy based on native ecological balance and the local income.

Considering that maternal–fetal toxicological studies play a fundamental role in understanding the risks associated with pregnancy and the neonatal consequences, studies dealing with these issues are highly relevant. Thus, the evaluation of the toxicological safety of *C. icaco* can guide on its use during pregnancy and depict this natural product as a new form of treatment by combining environmental, socioeconomic, and innovation aspects. Given this, this study was developed to analyze the embryotoxic and teratogenic effects of the oral administration of the aqueous extract of *C. icaco* during the pre-implantation and organogenesis periods in Wistar rats.

## 2. Materials and Methods

### 2.1. Plant Material

The leaves of *C. icaco* were collected from Itamaracá Island (7°46′39.29″ S/34°50′4.27″ W), which is located in the state of Pernambuco, Brazil, in July 2017. The authenticity of a species sample was confirmed by Dr. Leidiana Lima, from the Federal Rural University of

Pernambuco, who also deposited a voucher specimen of *C. icaco* L. at the herbarium under reference number 83131.

### 2.2. Plant Extract Preparation

To obtain the aqueous extract, the leaves were dried at 40 °C in an aerated stove and ground in knife mills. The powder (50 g) was subjected to extraction by infusion in distilled water (1000 mL) at 100 °C for 15 min. The extract was filtered, concentrated in a rotary evaporator under reduced pressure, and then lyophilized. The sample was stored at 4 °C and afterwards solubilized in saline at the desired concentrations (100, 200, and 400 mg/kg) minutes before the experiments were performed.

### 2.3. Animals

Ninety-day-old female rats (*Rattus norvegicus*) were acquired from the animal facility of the Department of Antibiotics at the Federal University of Pernambuco (UFPE). The rats were provided with water and LABINA Purina Brazil feed ad libitum and were maintained at a temperature of $22 \pm 2$ °C and humidity of $60 \pm 1\%$ levels and kept to a 12/12 light/dark cycle. The handling and procedures for the animals were authorized by the Ethical Committee for Animal Research of UFPE, Brazil (Protocol number 23076.016550/2016-38).

#### 2.3.1. Mating Period and Experimental Groups

To initiate the mating process according to OECD Guideline 421 [20], nulliparous females were placed in contact with adult males in a 2:1 ratio at the onset of the dark phase cycle. After 12 h (at the start of the light phase), vaginal lavage was performed using 0.9% NaCl for microscopic analysis. Mating was confirmed by the presence of sperm in the collected wash, which indicated the estrous phase of the estrous cycle and determined day 0 of pregnancy [21]. Mating continued until enough pregnant rats were obtained ($n = 80$). After identifying pregnancy, the rats were randomly assigned to eight experimental groups ($n = 10$/group), with four groups treated during the pre-implantation phase (day 0 to day 6 of gestation) and four treated during the organogenesis phase (6th day to 15th day of gestation). The rats received water as a control vehicle (5 mL/kg/day) or the aqueous extract *C. icaco* leaf (AECi) at doses of 100, 200, and 400 mg/kg/day, by gavage, at specific times. The selection of these doses was based on previous research studies [22].

#### 2.3.2. Analysis of Maternal Toxicity

Daily observations were made to assess clinical signs of toxicity, including but not limited to diarrhea, piloerection, stress, changes in motor activity, and bleeding. The feed and water consumption of the dams as well as their weight gain were also monitored [20].

#### 2.3.3. Evaluation of the Maternal Reproductive Performance
Pre-Implantation Period

On the 7th day of pregnancy, the rats in the pre-implantation group ($n = 40$) were intraperitoneally anesthetized (i.p.) with ketamine hydrochloride (60 mg/kg) and xylazine hydrochloride (6 mg/kg). Subsequently, a laparotomy was conducted to extract the uterine horns and assess the number of implantations. The ovaries were also retrieved and their weight was recorded, along with a count of their corpus luteum. The pre-implantation loss rate was calculated using the formula:

$$\text{Pre-implantation losses} = [(\text{Number of corpora lutea} - \text{number of implantations})/\text{Number of corpora lutea}] \times 100. \qquad (1)$$

The maternal organs (thymus, heart, lungs, liver, kidneys, pancreas, adrenals, and spleen) were collected, weighed, and evaluated for the presence of macroscopic changes. Relative weight was calculated as the ratio between organ weight and female body weight on the day of sacrifice $\times$ 100.

Organogenesis Period

On the 21st day of gestation, the rats were subjected to anesthesia as described in the Pre-Implantation Period section and subsequently underwent laparotomy. They were then euthanized by making an incision in the heart. Macroscopic examinations were conducted in the uterine horns to determine the viability of the fetuses and to identify any instances of early or late resorption. The implantation index, resorption rate, and post-implantation loss were calculated using the following formulas:

$$\text{Implantation index} = (\text{total number of implantation sites}/\text{total number of corpora lutea}) \times 100 \quad (2)$$

$$\text{Resorption index} = (\text{total number of resorption sites}/\text{total number of implantation sites}) \times 100 \quad (3)$$

$$\text{Post-implantation loss rate} = (\text{number of implants} - \text{number of live fetuses}/\text{number of implants}) \times 100 \quad (4)$$

Maternal organs (thymus, heart, lungs, liver, kidneys, pancreas, adrenals, and spleen) were collected, weighed, and analyzed for macroscopic malformations. After the hysterectomy, the fetuses were weighed and sexed by observing the anogenital distance (distance between the animal's genital area and anus): the female's distance is shorter than the male's. The placental index and the sex ratio were calculated according to the following equations:

$$\text{Placental index} = (\text{placental weight (g)}/\text{fetal weight (g)}) - 100 \quad (5)$$

$$\text{Sex ratio} = (\text{number of males}/\text{number of females}) \quad (6)$$

### 2.3.4. Visceral Analysis in Organogenesis

Half of each litter was fixed in a Bouin's solution (50 mL formaldehyde, 50 mL acetic acid, 752 mL 95% ethyl alcohol, and 148 mL distilled water) for one week for subsequent visceral examination using Wilson's [23] (1965) serial section method. Detailed analyses of the sections of the head, thorax, and pelvis were performed.

### 2.3.5. Skeletal Analysis in Organogenesis

After conducting external examinations of the fetuses, the remaining half of each litter was immersed in 70% alcohol for 24 h, followed by acetone for another 24 h, and then eviscerated and stained with alizarin red (0.5 mg) in 200 mL of 1% KOH (*w/v*) for the purpose of investigating skeletal changes using a method adapted from Staples and Schenell [24] (1964). The number, shape, and position of bones were examined to confirm fetal abnormalities and malformations based on the criteria proposed by Solecki et al. [25] (2001), and the counting and analysis of ossification points were performed according to Aliverti et al. [26] (1979).

### 2.3.6. Histological and Morphometric Analysis

Following laparotomy, the implantation sites were extracted and preserved in 10% (*v/v*) formalin for 24 h. Subsequently, the specimens were rinsed with running water and soaked in 70% alcohol (*v/v*) until histological examination. Histological analysis of the implantation sites was carried out with the aid of an Olympus® Bx50 microscope (Olympus Corporation, Tokyo, Japan), coupled with a Sony® video camera (Sony Group Corporation, Tokyo, Japan).

To perform the morphometric analysis of the placenta, two slides from each group were utilized and the placental disc regions were examined. Only the labyrinth, trophospongium, and trophoblastic giant cell regions were measured. The images were captured with a Sony® Video camera (Sony Group Cortoration, Tokyo, Japan) attached to the Olympus® Bx50 microscope (Olympus Corporation, Tokyo, Japan). The line morphometry application, calibrated in micrometers, was employed to measure the total area of the placental disc and its layers. The Optimas® 6.2 program for Windows was utilized for the morphometry. To

quantify the constituent elements of the placental disc layers, a 110-point graticule was used to identify the structures and cells, which were classified as (1) maternal vascularization; (2) small trophoblastic cells, undifferentiated (in contact with fetal vessels); (3) intermediate cells (in contact with maternal vessels); (4) giant trophoblastic (binucleated) cells; (5) syncytial cells (near the trophoblast region); and (6) mesenchyme, in the trophospongial region. In the labyrinth region, they were classified as (1) syncytial trophoblast; (2) fetal vessel wall; (3) lumen of fetal vessels; and (4) labyrinthine vascular bed (maternal vessel), and were analyzed with the 40× objective, with 10 fields randomly chosen per slide, and with three repetitions, making a total of 1100 points [27,28].

### 2.4. Statistical Analysis

Data normality was analyzed using the Shapiro–Wilk test and the results were expressed as mean ± standard deviation. Differences between groups were determined using analysis of variance (ANOVA), followed by Tukey's test or a Kruskal–Wallis test, followed by Dunn's post-test when applicable. Morphometric data were analyzed using the Kruskal–Wallis test, where the means were compared using the Wilcoxon–Mann–Whitney test. Statistical analysis was performed using the Graph Pad Prism 6.0 software and the significance level was set at $p < 0.05$.

## 3. Results

### 3.1. Maternal Toxicity

Table 1 shows that the treatment with AECi during the period of embryogenesis and organogenesis did not affect the body weight and feed consumption of the pregnant rats; also, water consumption was not affected. Additionally, no clinical signs of toxicity, such as piloerection, diarrhea, salivation, bleeding, or death were observed.

**Table 1.** Effect of the aqueous extract of *C. icaco* leaves, orally administrated, on the body weight of female Wistar rats during the pre-implantation period and organogenesis.

| | Pre-Implantation | |
|---|---|---|
| | Day 0 | Day 6 |
| Control | 200.03 ± 8.04 [a] | 210.50 ± 9.19 [a] |
| AECi 100 mg/kg | 199.25 ± 8.70 [a] | 201.50 ± 11.01 [a] |
| AECi 200 mg/kg | 202.18 ± 7.00 [a] | 209.26 ± 13.04 [a] |
| AECi 400 mg/kg | 199.47 ± 6.51 [a] | 204.00 ± 6.23 [a] |
| | Organogenesis | |
| | Day 6 | Day 21 |
| Control | 209.14 ± 5.00 [a] | 305.30 ± 10.15 [a] |
| AECi 100 mg/kg | 207.50 ± 10.60 [a] | 310.5 ± 10.70 [a] |
| AECi 200 mg/kg | 219.00 ± 15.88 [a] | 314.75 ± 15.02 [a] |
| AECi 400 mg/kg | 212.50 ± 14.97 [a] | 305.60 ± 11.00 [a] |

Means followed by the same letter in the lines do not differ significantly from each other using the ANOVA test followed by Tukey's test. Values represent the mean ± standard deviation of the mean (*n* = 10/group).

Absolute and relative organ weights of the pregnant rats treated with AECi during the pre-implantation and organogenesis period (Tables 2 and 3, respectively) did not differ from the control animals; no macroscopic signs (texture and color) of toxicity were observed in the organs of the animals treated with the extract.

### 3.2. Reproductive Parameters of the Pre-Implantation Period

No differences were observed in the number of corpora lutea, number of implantation sites, or pre-implantation losses of rats treated with the *C. icaco* extract in the pre-implantation period when compared to the control group.

**Table 2.** Effect of the aqueous extract of *C. icaco* leaves, orally administrated, on the absolute (g) and relative (%) organ weights of Wistar rats during the pre-implantation period.

| Organs | Control | AECi | | |
|---|---|---|---|---|
| | | 100 mg/kg | 200 mg/kg | 400 mg/kg |
| Liver (g) | 11.22 ± 2.07 [a] | 11.36 ± 1.43 [a] | 11.09 ± 0.80 [a] | 11.96 ± 1.82 [a] |
| Liver (%) | 4.42 ± 0.62 [a] | 3.67 ± 0.27 [a] | 3.83 ± 0.28 [a] | 3.58 ± 0.32 [a] |
| Kidney (g) | 1.54 ± 0.15 [a] | 1.47 ± 0.22 [a] | 1.54 ± 0.21 [a] | 1.44 ± 0.16 [a] |
| Kidney (%) | 0.54 ± 0.05 [a] | 0.49 ± 0.04 [a] | 0.49 ± 0.06 [a] | 0.49 ± 0.03 [a] |
| Spleen (g) | 0.82 ± 0.11 [a] | 0.72 ± 0.10 [a] | 0.79 ± 0.10 [a] | 0.79 ± 0.09 [a] |
| Spleen (%) | 0.23 ± 0.30 [a] | 0.22 ± 0.03 [a] | 0.20 ± 0.05 [a] | 0.24 ± 0.02 [a] |
| Heart (g) | 0.78 ± 0.08 [a] | 0.77 ± 0.15 [a] | 0.77 ± 0.08 [a] | 0.76 ± 0.07 [a] |
| Heart (%) | 0.28 ± 0.04 [a] | 0.24 ± 0.00 [a] | 0.25 ± 0.02 [a] | 0.26 ± 0.01 [a] |
| Lungs (g) | 1.17 ± 0.17 [a] | 1.38 ± 0.15 [a] | 1.46 ± 0.18 [a] | 1.44 ± 0.16 [a] |
| Lungs (%) | 0.51 ± 0.08 [a] | 0.53 ± 0.22 [a] | 0.49 ± 0.03 [a] | 0.46 ± 0.05 [a] |
| Pancreas (g) | 0.72 ± 0.13 [a] | 0.77 ± 0.15 [a] | 0.74 ± 0.09 [a] | 0.75 ± 0.17 [a] |
| Pancreas (%) | 0.26 ± 0.02 [a] | 0.24 ± 0.01 [a] | 0.22 ± 0.02 [a] | 0.23 ± 0.02 [a] |
| Adrenals (g) | 0.08 ± 0.00 [a] | 0.07 ± 0.01 [a] | 0.07 ± 0.00 [a] | 0.07 ± 0.00 [a] |
| Adrenals (%) | 0.03 ± 0.00 [a] | 0.02 ± 0.00 [a] | 0.02 ± 0.00 [a] | 0.02 ± 0.00 [a] |
| Thymus (g) | 0.36 ± 0.03 [a] | 0.32 ± 0.02 [a] | 0.33 ± 0.03 [a] | 0.31 ± 0.01 [a] |
| Thymus (%) | 0.36 ± 0.03 [a] | 0.11 ± 0.02 [a] | 0.10 ± 0.03 [a] | 0.10 ± 0.02 [a] |
| Ovary (g) | 0.08 ± 0.00 [a] | 0.07 ± 0.01 [a] | 0.07 ± 0.00 [a] | 0.07 ± 0.00 [a] |
| Ovary (%) | 0.04 ± 0.01 [a] | 0.03 ± 0.00 [a] | 0.03 ± 0.01 [a] | 0.03 ± 0.00 [a] |
| Uterus (g) | 0.94 ± 0.15 [a] | 0.94 ± 0.12 [a] | 0.95 ± 0.08 [a] | 0.91 ± 0.14 [a] |
| Uterus (%) | 0.46 ± 0.02 [a] | 0.46 ± 0.01 [a] | 0.47 ± 0.03 [a] | 0.45 ± 0.01 [a] |

Means followed by the same letter in the lines do not differ significantly from each other using the ANOVA test followed by Tukey's test. Values represent the mean ± standard deviation of the mean ($n$ = 10/group). Relative weight was calculated as the ratio between organ weight and female body weight on the day of sacrifice × 100.

**Table 3.** Effect of the aqueous extract of *C. icaco* leaves, orally administrated, on the absolute (g) and relative (%) organ weights of Wistar rats treated during the period of organogenesis.

| Organs | Control | AECi | | |
|---|---|---|---|---|
| | | 100 mg/kg | 200 mg/kg | 400 mg/kg |
| Liver (g) | 12.98 ± 1.37 [a] | 11.17 ± 0.77 [a] | 12.36 ± 0.94 [a] | 11.06 ± 1.26 [a] |
| Liver (%) | 4.42 ± 0.62 [a] | 3.67 ± 0.27 [a] | 3.83 ± 0.28 [a] | 3.58 ± 0.32 [a] |
| Kidney (g) | 1.60 ± 0.19 [a] | 1.51 ± 0.14 [a] | 1.58 ± 0.10 [a] | 1.33 ± 0.52 [a] |
| Kidney (%) | 0.54 ± 0.05 [a] | 0.49 ± 0.04 [a] | 0.49 ± 0.06 [a] | 0.49 ± 0.03 [a] |
| Spleen (g) | 0.70 ± 0.09 [a] | 0.68 ± 0.12 [a] | 0.65 ± 0.24 [a] | 0.76 ± 0.10 [a] |
| Spleen (%) | 0.23 ± 0.30 [a] | 0.22 ± 0.03 [a] | 0.20 ± 0.05 [a] | 0.24 ± 0.02 [a] |
| Heart (g) | 0.84 ± 0.18 [a] | 0.74 ± 0.03 [a] | 0.83 ± 0.08 [a] | 0.80 ± 0.06 [a] |
| Heart (%) | 0.28 ± 0.04 [a] | 0.24 ± 0.00 [a] | 0.25 ± 0.02 [a] | 0.26 ± 0.01 [a] |
| Lungs (g) | 1.53 ± 0.27 [a] | 1.62 ± 0.22 [a] | 1.61 ± 0.10 [a] | 1.22 ± 0.54 [a] |
| Lungs (%) | 0.51 ± 0.08 [a] | 0.53 ± 0.22 [a] | 0.49 ± 0.03 [a] | 0.46 ± 0.05 [a] |
| Pancreas (g) | 0.78 ± 0.15 [a] | 0.63 ± 0.17 [a] | 0.68 ± 0.10 [a] | 0.71 ± 0.12 [a] |
| Pancreas (%) | 0.24 ± 0.01 [a] | 0.22 ± 0.01 [a] | 0.20 ± 0.01 [a] | 0.24 ± 0.02 [a] |
| Adrenals (g) | 0.09 ± 0.01 [a] | 0.09 ± 0.00 [a] | 0.08 ± 0.00 [a] | 0.08 ± 0.00 [a] |
| Adrenals (%) | 0.03 ± 0.00 [a] | 0.02 ± 0.00 [a] | 0.02 ± 0.00 [a] | 0.02 ± 0.00 [a] |
| Thymus (g) | 0.37 ± 0.03 [a] | 0.37 ± 0.03 [a] | 0.35 ± 0.03 [a] | 0.31 ± 0.05 [a] |
| Thymus (%) | 0.10 ± 0.01 [a] | 0.11 ± 0.00 [a] | 0.11 ± 0.01 [a] | 0.10 ± 0.01 [a] |

Means followed by the same letter in the lines do not differ significantly from each other using the ANOVA followed by Tukey's test. Values represent the mean ± standard deviation of the mean ($n$ = 10/group). Relative weight was calculated as the ratio between organ weight and female body weight on the day of sacrifice × 100.

### 3.3. Histopathology of Implantation Sites

The implantation sites of all experimental groups demonstrated good preservation appearance and were fully inserted into the uterine wall, with good development (Figure 1A–G). It was also possible to find evidence of cytotrophoblasts and several trophoblasts in mitosis (Figure 1B–H).

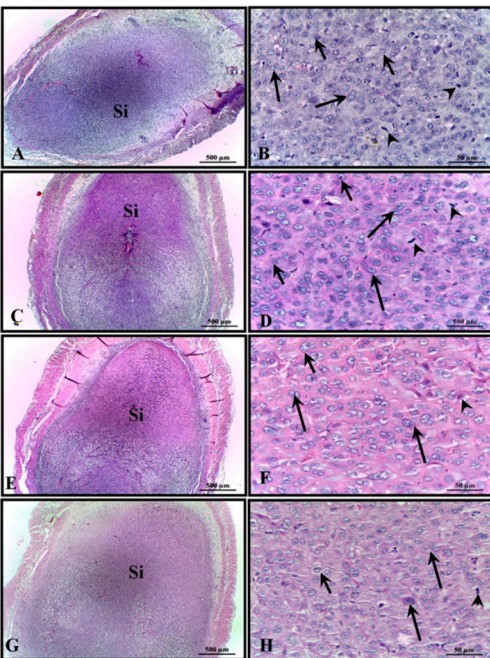

**Figure 1.** Micrograph of part of the uterus of rats showing the implantation sites (Si) of all of the animals. (**A**,**B**) control; (**C**,**D**) 100 mg/kg; (**E**,**F**) 200 mg/kg; and (**G**,**H**) 400 mg/kg. H&E stain. Long arrow: cytotrophoblast; arrowhead: trophoblast in mitosis.

### 3.4. Reproductive Parameters of the Organogenesis Period

During organogenesis, it was observed that the treatment with AECi at different doses did not affect the gestational parameters compared to the control group (Table 4).

**Table 4.** Effect of the aqueous extract of *C. icaco* leaves, orally administrated, on the reproductive parameters of Wistar rats treated during the period of organogenesis.

| Reproductive Parameters | Control | AECi Treatment | | |
|---|---|---|---|---|
| | | 100 mg/kg | 200 mg/kg | 400 mg/kg |
| Pregnant rats | 10 | 10 | 10 | 10 |
| Number of live fetuses | 102 (100%) | 105 (100%) | 100 (100%) | 93 (100%) |
| Number of death fetuses | 0 | 0 | 0 | 0 |
| Number of implantations | 104 | 106 | 102 | 94 |
| Number of resorption sites | 2 | 1 | 2 | 1 |
| Number of corpora lutea [b] | 10.5 ± 1.84 [a] | 10.6 ± 1.26 [a] | 10.5 ± 1.22 [a] | 9.5 ± 1.35 [a] |
| Implantation index (%) [c] | 99.04 [a] | 100 [a] | 98.07 [a] | 98.94 [a] |
| Resorption index (%) [c] | 1.92 [a] | 0.94 [a] | 1.96 [a] | 1.06 [a] |
| Post-implantation loss rate (%) [b] | 1.96 [a] | 0.94 [a] | 1.96 [a] | 1.09 [a] |
| Ovaries absolute weight (g) [b] | 0.14 ± 0.04 [a] | 0.15 ± 0.01 [a] | 0.14 ± 0.01 [a] | 0.14 ± 0.03 [a] |
| Ovaries relative weight (%) [b] | 0.04 ± 0.01 [a] | 0.05 ± 0.00 [a] | 0.04 ± 0.00 [a] | 0.04 ± 0.01 [a] |
| Male body weights (g) [b] | 5.54 ± 0.40 [a] | 5.46 ± 0.51 [a] | 5.34 ± 0.65 [a] | 5.43 ± 0.44 [a] |
| Female body weights (g) [b] | 5.20 ± 0.44 [a] | 5.20 ± 0.57 [a] | 5.37 ± 0.36 [a] | 5.12 ± 0.48 [a] |
| Placental weight [b] | 4.99 ± 0.71 [a] | 5.26 ± 0.68 [a] | 5.06 ± 0.64 [a] | 4.50 ± 0.59 [a] |

Values followed by the same letter in the lines do not differ significantly from each other using the ANOVA test followed by Tukey's test (b) and the Kruskal–Wallis test, followed by the Dunn's (c) post test. Implantation index (total number of implantation sites/total number of corpora lutea × 100); resorption index (total number of resorption sites/total number of implantation sites × 100); post-implantation loss rate (number of implants-number of live fetuses/number of implants × 100); and placental index (placental weight (g)/fetal weight (g) × 100). Values are expressed as mean ± SEM (a) or median (b). Statistical analyses were performed using ANOVA, followed by a Tukey's test (parametric data) and Kruskal–Wallis test, followed by Dunn's post test (nonparametric data).

### 3.5. Teratogenicity

The administration of AECi did not cause malformations or anomalies (external or internal) in the viscera of the analyzed neonatals. In the differential analysis of skeletal anomalies and malformations, it was possible to evidence some alterations; for example, agenesis in the anterior and posterior proximal phalanges was observed in the groups treated with 100 and 400 mg/kg of AECi, and agenesis of the caudal vertebrae was observed in all treated groups. Although diagnosed, these alterations did not present statistical differences when compared to the control group (Table 5).

**Table 5.** Effect of the extract of *C. icaco* leaves, orally administered, on neonatal skeletal changes in pregnant Wistar rats treated during the period of organogenesis.

| Skeletal Malformations | Control | AECi | | |
|---|---|---|---|---|
| | | 100 mg/kg | 200 mg/kg | 400 mg/kg |
| Agenesis of anterior phalanges | 1/51 (1.78%) | 1/53 (1.88%) | 0/50 (0%) | 2/47 (4.2%) |
| Agenesis of posterior phalanges | 2/51 (3.57%) | 2/53 (3.77%) | 1/50 (2%) | 2/47 (4.5%) |
| Cervical vertebrae agenesis | 0/51 (0%) | 0/53 (0%) | 0/50 (0%) | 0/47 (0%) |
| Caudal certebrae agenesis | 2/51 (3.57%) | 3/53 (5.66%) | 2/50 (4%) | 2/47 (4.5%) |
| Sternebrium agenesis | 0/51 (0%) | 0/53 (0%) | 0/50 (0%) | 0/47 (0%) |
| Absence of xiphoid process | 0/51 (0%) | 0/53 (0%) | 0/50 (0%) | 0/47 (0%) |
| Enlarged fontanelle | 0/51 (0%) | 0/53 (0%) | 0/50 (0%) | 0/47 (0%) |
| Wavy ribs | 0/51 (0%) | 0/53 (0%) | 0/50 (0%) | 0/47 (0%) |

The results of fetal skeletal malformations were organized based on the ratio between the number of affected fetuses/total number of analyzed fetuses. Values in parentheses represent the percentage of observed skeletal malformations. Statistical differences between the control and treated groups were calculated using Pearson's chi-square test.

The values of the ossification points of the fetuses from pregnant rats treated with AECi were similar to those of the control group (these values were in accordance with the values for the species).

### 3.6. Placental Histopathology

In all experimental groups, the placental disc was well developed, with a very characteristic labyrinth, trophospongium, and giant trophoblastic cell layers. In the labyrinth layer, the outermost and thickest region, numerous gaps were found containing maternal and fetal vessels. In the trophosponge layer, undifferentiated trophoblasts were observed. The last layer is formed of giant trophoblastic cells, which mix with the decidua (Figure 2A–D).

### 3.7. Placental Morphometric Analysis

The morphometric analysis demonstrated a reduced ($p < 0.05$) area in the total placental region for animals treated with 400 mg/kg (254.21 $\mu m^2$) when compared to the control group (309.32 $\mu m^2$). Furthermore, only the group that received the highest dose of AECi showed a significant reduction in the total area of the placental disc layers—without, however, differing in the quantification of its constituents (Tables 6–8).

**Table 6.** Effect of the aqueous extract of *C. icaco* leaves, orally administered, on the measurement of the areas ($\mu m^2$) of placental disc layers in experimental groups.

| Layers | Control | AECi | | |
|---|---|---|---|---|
| | | 100 mg/kg | 200 mg/kg | 400 mg/kg |
| Labyrinth | 133.50 ± 7.14 [a] | 130.77 ± 5.98 [a] | 129.32 ± 8.69 [a] | 101.73 ± 5.12 [b] |
| Trophospongium | 91.09 ± 4.45 [a] | 89.37 ± 3.05 [a] | 90.82 ± 5.15 [a] | 68.19 ± 6.76 [b] |
| Giant Trophoblastic Cells | 23.15 ± 1.17 [a] | 22.60 ± 2.10 [a] | 22.16 ± 2.84 [a] | 18.93 ± 1.00 [b] |

Means followed by the same letter in the lines do not differ significantly from each other using the Wilcoxon–Mann–Whitney ($p < 0.05$) test. Values represent the mean ± standard deviation.

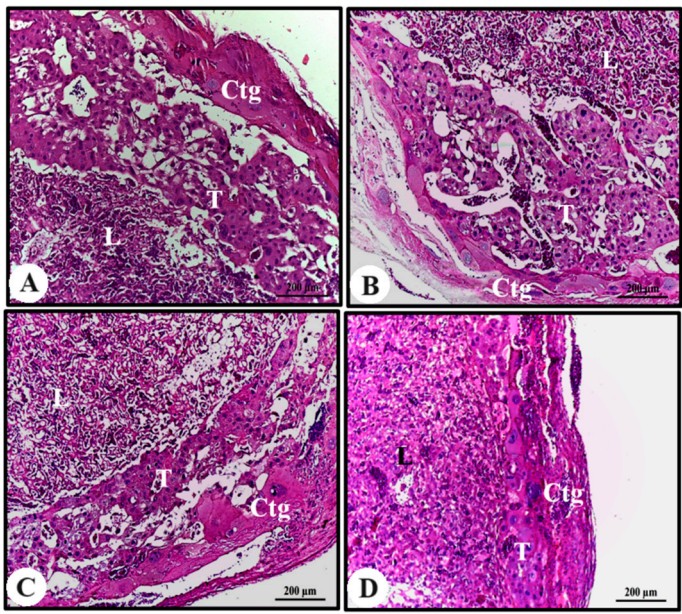

**Figure 2.** Photomicrography of the placentas of animals in the experimental groups. (**A**) Control; (**B**) AECi 100 mg/kg; (**C**) AECi 200 mg/kg; and (**D**) AECi 400 mg/kg. H&E stain. Ctg: giant trophoblastic cells; T: trophoblastic; and L: labyrinth.

**Table 7.** Effect of the aqueous extract of *C. icaco* leaves, orally administrated, on the counting of elements in the labyrinth region of the placental disc of animals in the experimental groups.

| Layers | Control | AECi | | |
|---|---|---|---|---|
| | | 100 mg/kg | 200 mg/kg | 400 mg/kg |
| Syncytial trophoblast | 16.20 ± 4.92 [a] | 14.53 ± 5.34 [a] | 12.47 ± 6.61 [a] | 11.06 ± 3.31 [a] |
| Fetal vessel wall | 21.1 ± 5.30 [a] | 19.45 ± 1.69 [a] | 20.12 ± 3.14 [a] | 18.23 ± 5.81 [a] |
| Fetal vessel lumen | 46.3 ± 13.26 [a] | 42.22 ± 11.87 [a] | 39.68 ± 12.01 [a] | 48.26 ± 3.16 [a] |
| Maternal vessel | 16.6 ± 7.89 [a] | 15.43 ± 9.17 [a] | 14.98 ± 7.77 [a] | 17.36 ± 6.95 [a] |

Means followed by the same letter in the lines do not differ significantly from each other using the Wilcoxon–Mann–Whitney ($p < 0.05$) test. Values represent the mean ± standard deviation.

**Table 8.** Effect of the aqueous extract of *C. icaco* leaves, orally administrated, on the counting of elements in the trophospongium region of the placental disc.

| Layers | Control | AECi | | |
|---|---|---|---|---|
| | | 100 mg/kg | 200 mg/kg | 400 mg/kg |
| Maternal vascularization | 14.43 ± 3.65 [a] | 13.98 ± 2.02 [a] | 11.54 ± 3.21 [a] | 15.39 ± 2.76 [a] |
| Undifferentiated trophoblastic cells | 8.11 ± 1.03 [a] | 9.44 ± 1.72 [a] | 7.48 ± 2.90 [a] | 8.31 ± 2.12 [a] |
| Intermediate cells | 5.30 ± 1.20 [a] | 3.18 ± 1.57 [a] | 4.47 ± 2.65 [a] | 4.07 ± 3.83 [a] |
| Giant trophoblastic cells | 1.62 ± 0.19 [a] | 1.13 ± 0.14 [a] | 0.99 ± 1.95 [a] | 1.01 ± 0.75 [a] |
| Syncytial cells | 0.52 ± 0.06 [a] | 0.49 ± 0.17 [a] | 0.50 ± 0.11 [a] | 0.50 ± 0.13 [a] |
| Mesenchyme | 75.34 ± 9.33 [a] | 70.69 ± 6.66 [a] | 73.71 ± 4.49 [a] | 71.03 ± 4.80 [a] |

Means followed by the same letter in the lines do not differ significantly from each other using the Wilcoxon–Mann–Whitney ($p < 0.05$) test. Values represent the mean ± standard deviation.

## 4. Discussion

The use of natural products for different disease treatments has increased over the years; furthermore, studies on the effects of these products on the reproductive performance of females is of fundamental importance [29]. The *C. icaco* species is widely used in traditional medicine as an alternative and natural treatment of diseases [30,31]; however, to the best of our knowledge, no studies have been found reporting on the safety of its species during the gestational period. Therefore, several reproductive parameters were evaluated

after the administration of the aqueous extract of *C. icaco* leaves to obtain more information about its reproductive toxicity.

In previous studies carried out by our research group, the phytochemical analysis performed by UPLC-DAD-ESI-QTOF-MS/MS identified the presence of terpenes, aglycones, and glycosylated flavonoids derived from myricetin and quercetin in the aqueous extract of *C. icaco* leaves (AECi) [19]. Phenolic compounds are substances included in the group of phytoestrogens capable of performing estrogen-like activities. Some phytoestrogens can cross the placental barrier and promote harmful effects on neonates [32,33]. At high concentrations, phytoestrogens have anti-estrogenic activity, and decreased estrogen levels can affect the development of placental cells [34–36].

Rodents are excellent models for the study of reproductive physiology due to their small size, high reproduction rate, and the ease of obtaining inbred lines [37]. The general health of females in the gestational period provides relevant information about the reproductive toxic effects of a substance [38]. Behavioral changes, variations in food consumption, and changes in body and organ weights during this phase can be used as an important indicator of systemic toxicity [39]. Our results showed that there was no behavioral change in rats treated with AECi when compared to the control group. Furthermore, no changes were observed in food consumption, water consumption, and body or organ weights, indicating that the administration of different doses of AECi (100, 200, and 400 mg/kg) does not cause systemic maternal toxicity. Ribeiro et al. [19] (2020) showed that the aqueous extract of *C. icaco* presented low toxicity as the single administration of a 2000 mg/Kg dose did not affect the intake of water and food or body weight of the treated animals. These authors also found that animals treated with AECi at doses of 100, 200, and 400 mg/kg for 28 consecutive days showed no death or signs of toxicity (pilocration, diarrhea, or changes in locomotor activity).

In female rats, the process of implantation of the blastocyst in the endometrium occurs from the 1st to the 5th day of gestation and, during this period, the embryo is quite susceptible to lethality [40]. The success of pregnancy depends on the maintenance of progesterone and estrogen levels. Estrogens produce a suitable environment for fertilization, implantation, and nutrition, while progesterone is responsible for maintaining the conditions of the endometrium for a possible pregnancy. The corpus luteum is the primary source of these hormones, which are very important in the process of preparing the uterine mucosa to receive the embryo [37,41].

Successful implantation is closely related to endometrial cell proliferation, decidualization, and increased blood flow [42,43]. Various xenobiotics can interfere with mitotic division, thus interrupting the pregnancy. These substances can interfere before and after the implantation process, resulting in pre- and post-implantation embryo loss, developmental delay, malformations, and even fetal death [44,45]. In the literature, it is possible to find several toxic plants that affect the fertilization and implantation process, such as *Coleus barbatus*, *Jatropha curcas*, and *Indigofera suffruticosa* [46–48]. In the present study, it is possible to affirm that the treatment with AECi did not change the number of corpora lutea and the number of implantation sites when compared to the control group. These data revealed the absence of embryotoxic effects of AECi, suggesting a normal developmental capacity of implanted blastocysts.

The study of the placenta is an essential tool to assess placental toxicity and its subsequent effects on the fetus. The adequate supply of nutrients through the placenta is directly linked to fetal development in the uterine environment, as any placental dysfunction promotes a higher incidence of fetal distress [49,50]; given this, parameters such as fetal weight, placental weight, and placental index feature in the evaluation of embryofetotoxicity. In this present study, it was observed that fetal body weight, placental weight, and placental index were not affected by the administration of AECi, thus confirming the extract safety for the fetus.

Rat placenta is hemochorial (discoidal) and divided into a fetal part composed of the labyrinth and junctional zone (basal zone), and a maternal part, in this case composed of the

decidua and metrial gland. Maternal–fetal exchanges of nutrients, metabolites, and gases occur in the labyrinth and, in this region, a layer of trophoblasts separates the maternal from the fetal blood. The basal zone, which forms below the labyrinth, is composed of cells containing glycogen, spongiotrophoblasts, and giant trophoblastic cells; it plays an important role in the metabolism of chemicals. The decidua, the maternal portion, consists of decidual cells of the mesometrium and the metrial gland, formed by fibroblasts, trophoblasts, fibroblasts, and natural killer cells [51–53].

Despite the reduction in the total area of the placental disc of the animals treated with 400 mg/kg of AECi and that observed in their layers, no difference was observed in its constituents. It is possible that flavonoids exert an antiestrogenic role and thereby interfered with the total area of the placenta and placental constituents.

The absence of changes in both the ossification points and the viscera of animals treated with AECi suggests that the extract does not possess teratogenic effects. As a future perspective, we will consider the elucidation of the results reported in the topic "Placental morphometric analysis" of this work.

## 5. Conclusions

Based on the results, we can affirm that the oral administration of the aqueous extract of *C. icaco* (AECi) at different doses (100, 200, and 400 mg/kg) did not affect body weight and water and feed consumption of pregnant rats during the period of embryogenesis and organogenesis. Additionally, no clinical signs of toxicity (piloerection, diarrhea, salivation, bleeding, and death) were observed. After euthanasia, the absolute and relative organ weights of the treated rats were like the control ones; neither presented macroscopic signs of toxicity in the organs. Results were also similar for treated and control animals, considering the periods of pre-implantation and organogenesis. Finally, the administration of AECi did not cause malformations or anomalies in the viscera of neonates from treated rats. There is a lack of research on studies reporting *C. icaco* toxicity, during the gestational period or any other. Thus, it is possible to affirm that AECi did not induce any changes in maternal toxicity and reproductive performance, highlighting the importance of this investigation and suggesting that, under the experimental conditions used in this study, the extract does not cause reproductive toxicity.

**Author Contributions:** Conceptualization, T.G.d.S.; Data curation, P.B.S.d.A., F.M.R.d.S.J. and J.B.N.F.S.; Formal analysis, N.E.R.R., J.B.N.F.S. and Á.A.C.T.; Funding acquisition, T.G.d.S.; Investigation, N.E.R.R. and A.G.W.; Methodology, N.E.R.R., A.R.d.S.O., D.M.N. and Á.A.C.T.; Project administration, T.G.d.S.; Supervision, T.G.d.S.; Visualization, M.d.G.C.d.C. and F.M.R.d.S.J.; Writing—original draft, N.E.R.R., S.M.d.A.L. and P.B.S.d.A.; Writing—review and editing, M.d.G.C.d.C. All authors have read and agreed to the published version of the manuscript.

**Funding:** This study was supported by National Council for Scientific and Technological Development (CNPq)—Number 312087/2019-5.

**Institutional Review Board Statement:** The animal study protocol was approved by the Ethics Committee for Animal Research of the Federal University of Pernambuco (UFPE), Brazil (23076.016550/2016-38, 30 August 2016).

**Informed Consent Statement:** Not applicable.

**Data Availability Statement:** Not applicable.

**Acknowledgments:** The authors are grateful to the National Council for Scientific and Technological Development (CNPq) for the financial support.

**Conflicts of Interest:** The authors declare no conflict of interest.

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
