# Peer review of "Effect of the Aqueous Extract of Chrysobalanus icaco Leaves on Maternal Reproductive Outcomes and Fetal Development in Wistar Rats"

_cimb, doi:10.3390/cimb45090479_

Round 1

Reviewer 1 Report

This original article addresses a subject of interest: the safety of medicinal plants, adding knowledge to the toxicological profile of Chrysobalanus icaco. The paper presents the harmonious continuation of the previous research on the toxicity of the mentioned species.

The article deserves to be published in CIMB, however, there are some points in which improvements should be made to strengthen the manuscript.

            It would be beneficial for the reader to add a paragraph to the introduction enumerating the various types of toxicity studies and their timeline.

General observations:

- I would suggest separating the title of the tables from the explanatory notes.

- Please check that the name of the species is written in italics throughout the manuscript.

Table 2 – please remove the word “leaves” located towards the end of row 206.

Figure 1 – please add “Si” to the legend.

Line 331 – please mention the names of the species.

Please add to the discussion section some comments linking the current results to the ones published by your team in 2020. Also, a perspectives paragraph would be interesting.

The conclusions are consistent with the evidence and arguments presented.

The references are appropriate.

Please check line 103 - "whith to".

Author Response

Dear Editor,

       I am pleased to send you the revised manuscript with corrections marked in red. Please find below our responses, point by point, to the reviewer’s questions and a description of the changes introduced into the Manuscript.

        We are grateful to the reviewers of the Current Issues in Molecular Biology Journal for their valuable and constructive comments, which contributed to improving the quality of this manuscript. We also hope some misunderstandings were clarified and that our response to the comments of the reviewers is satisfactory, thus allowing the manuscript to be considered for publication now in this important journal.

          Please do not hesitate to contact me if you have any further questions.

My best regards,

Comments and Suggestions for Reviewer 1

This original article addresses a subject of interest: the safety of medicinal plants, adding knowledge to the toxicological profile of Chrysobalanus icaco. The paper presents the harmonious continuation of the previous research on the toxicity of the mentioned species.

The article deserves to be published in CIMB, however, there are some points in which improvements should be made to strengthen the manuscript.

Comment 1: It would be beneficial for the reader to add a paragraph to the introduction enumerating the various types of toxicity studies and their timeline.

Response 1: Dear reviewer, thank you for your valuable comment. We added the requested information into the revised manuscript.

General observations:

Comment 2: I would suggest separating the title of the tables from the explanatory notes.

Response 2: Dear reviewer, thank you for your valuable comment. Modifications have been made as suggested.

Comment 3: Please check that the name of the species is written in italics throughout the manuscript.

Response 3: Dear reviewer, thank you for your valuable comment. The modifications have been made as suggested.

Comment 4: Table 2 – please remove the word “leaves” located towards the end of row 206.

Response 4: Dear reviewer, thank you for your valuable comment. The correction was made.

Comment 5: Figure 1 – please add “Si” to the legend.

Response 5: Dear reviewer, thank you for your valuable comment.  “Si” was added in the legend.

Comment 6: Line 331 – please mention the names of the species.

Response 6: Dear reviewer, thank you for your valuable comment. We added the name of the species as requentes; please ser limes 351-352 of the revised manuscript.

Comment 7: Please add to the discussion section some comments linking the current results to the ones published by your team in 2020. Also, a perspectives paragraph would be interesting.

Response 7: Dear reviewer, thank you for your valuable comment. We added the requested information into the revised manuscript.

Comment 8: The conclusions are consistent with the evidence and arguments presented.

Response 8: Dear reviewer, thank you for your valuable comment.

Comment 9: The references are appropriate.

Response 9: Dear reviewer, thank you for your valuable comment.

Reviewer 2 Report

I did not find any major concerns. It would be better to add details of the Authentication of the plant sample details/pictures (botanical characters) Details on the standardization of the extract (at least some basic assessment for repeatability of the study in terms of total phenolics, tannins, flavonoids or a HPLC chromatogram). These would be necessary for anybody to repeat the study and also for ascertaining the results obtained in this study.

Language is fine.

Author Response

Dear Editor,

        I am pleased to send you the revised manuscript with corrections marked in red. Please find below our responses, point by point, to the reviewer’s questions and a description of the changes introduced into the Manuscript.

        We are grateful to the reviewers of the Current Issues in Molecular Biology Journal for their valuable and constructive comments, which contributed to improving the quality of this manuscript. We also hope some misunderstandings were clarified and that our response to the comments of the reviewers is satisfactory, thus allowing the manuscript to be considered for publication now in this important journal.

         Please do not hesitate to contact me if you have any further questions.

My best regards,

Comments and Suggestions for Reviewer 2

Comment: I did not find any major concerns. It would be better to add details of the Authentication of the plant sample details/pictures (botanical characters) Details on the standardization of the extract (at least some basic assessment for repeatability of the study in terms of total phenolics, tannins, flavonoids or a HPLC chromatogram). These would be necessary for anybody to repeat the study and also for ascertaining the results obtained in this study.

Response: Dear reviewer, thank you for your valuable comments. Information regarding the botanical characteristics of the species was added to the text and can be found in lines 57-59. Unfortunately, it was not possible to add photos of the plant, as we lost the file with the photos.

Considering your suggestion, I affirm that the bioactive composition analysis was already published in the manuscript entitled "Acute and repeated dose 28-day oral toxicity of Chrysobalanus icaco L. leaf aqueous extract" and is cited in this manuscript (Reference nº 19).

Reviewer 3 Report

The manuscript covers the very important issue of the toxicity of aqueous extracts (infusions) of medicinal plants. Interest in preparations from plants for the treatment of a wide range of conditions has been growing rapidly in recent years, mainly because of the belief that they are devoid of the toxic effects that accompany synthetic drugs. This is not entirely true, so undertaking a study to evaluate the toxic effects of aqueous extracts of Chrysobalanus icaco leaves on maternal reproductive outcomes and fetal development in rats is a good step in the right direction. The need for this research is further supported by the fact that there is a lack of information in the World literature indicating the safety of Chrysobalanus icaco extracts during the gestational period. However, extensive studies conducted on Wistar rats have indicated that aqueous leaf extracts of the above-mentioned plant at doses of 100, 200 and 400 mg/kg are safe for use and do not show toxic effects on the reproductive performance of rats.

The paper was well designed and done, and the results obtained were presented in six clear Tables and two Figures. The discussion does not raise any objections. The literature (51 references) was carefully selected, the vast majority from the last 10 years. Only nine references were published below 2000, including one in 1965. Despite careful analysis of the text of the manuscript, I found no errors.

Author Response

Dear Editor,

        I am pleased to send you the revised manuscript with corrections marked in red. Please find below our responses, point by point, to the reviewer’s questions and a description of the changes introduced into the Manuscript.

        We are grateful to the reviewers of the Current Issues in Molecular Biology Journal for their valuable and constructive comments, which contributed to improving the quality of this manuscript. We also hope some misunderstandings were clarified and that our response to the comments of the reviewers is satisfactory, thus allowing the manuscript to be considered for publication now in this important journal.

            Please do not hesitate to contact me if you have any further questions.

My best regards,

Comments and Suggestions for Reviewer 3

Comments: The manuscript covers the very important issue of the toxicity of aqueous extracts (infusions) of medicinal plants. Interest in preparations from plants for the treatment of a wide range of conditions has been growing rapidly in recent years, mainly because of the belief that they are devoid of the toxic effects that accompany synthetic drugs. This is not entirely true, so undertaking a study to evaluate the toxic effects of aqueous extracts of Chrysobalanus icaco leaves on maternal reproductive outcomes and fetal development in rats is a good step in the right direction. The need for this research is further supported by the fact that there is a lack of information in the World literature indicating the safety of Chrysobalanus icaco extracts during the gestational period. However, extensive studies conducted on Wistar rats have indicated that aqueous leaf extracts of the above-mentioned plant at doses of 100, 200 and 400 mg/kg are safe for use and do not show toxic effects on the reproductive performance of rats.

The paper was well designed and done, and the results obtained were presented in six clear Tables and two Figures. The discussion does not raise any objections. The literature (51 references) was carefully selected, the vast majority from the last 10 years. Only nine references were published below 2000, including one in 1965. Despite careful analysis of the text of the manuscript, I found no errors.

Response: Dear reviewer, thank you for your valuable comments.